# Whole family-based physical activity promotion intervention: the Families Reporting Every Step to Health pilot randomised controlled trial protocol

Justin M Guagliano ![ORCID],[1] Helen Elizabeth Brown,[1] Emma Coombes,[2] Elizabeth S Haines,[1] Claire Hughes,[3] Andrew P Jones,[2] Katie L Morton,[1] Esther van Sluijs[1]

[1]MRC Epidemiology Unit and UKCRC Centre for Diet and Activity Research, University of Cambridge, Cambridge, UK
[2]Norwich Medical School and UKCRC Centre for Diet and Activity Research, University of East Anglia, Norwich, UK
[3]Centre for Family Research, University of Cambridge, Cambridge, UK

**Correspondence to**
Dr Justin M Guagliano;
justin.guagliano@mrc-epid.cam.ac.uk

## ABSTRACT

**Introduction** Family-based physical activity (PA) interventions present a promising avenue to promote children's activity; however, high-quality experimental research is lacking. This paper describes the protocol for the FRESH (Families Reporting Every Step to Health) pilot trial, a child-led family-based PA intervention delivered online.

**Methods and analysis** FRESH is a three-armed, parallel-group, randomised controlled pilot trial using a 1:1:1 allocation ratio with follow-up assessments at 8 and 52 weeks postbaseline. Families will be eligible if a minimum of one child in school Years 3–6 (aged 7–11 years) and at least one adult responsible for that child are willing to participate. Family members can take part in the intervention irrespective of their participation in the accompanying evaluation and vice versa. Following baseline assessment, families will be randomly allocated to one of three arms: (1) FRESH; (2) pedometer-only or (3) no-intervention control. All family members in the pedometer-only and FRESH arms receive pedometers and generic PA promotion information. FRESH families additionally receive access to the intervention website; allowing participants to select step challenges to 'travel' to target cities around the world, log steps and track progress as they virtually globetrot. Control families will receive no treatment. All family members will be eligible to participate in the evaluation with two follow-ups (8 and 52 weeks). Physical (eg, fitness and blood pressure), psychosocial (eg, social support) and behavioural (eg, objectively measured family PA) measures will be collected at each time point. At 8-week follow-up, a mixed methods process evaluation will be conducted (questionnaires and family focus groups) assessing acceptability of the intervention and evaluation. FRESH families' website engagement will also be explored.

**Ethics and dissemination** This study received ethical approval from the Ethics Committee for the School of the Humanities and Social Sciences at the University of Cambridge. Findings will be disseminated via peer-reviewed publications, conferences and to participating families.

**Trial registration number** ISRCTN12789422

### Strengths and limitations of this study

► This pilot trial is among the first physical activity (PA) interventions that specifically targets and evaluates whole family engagement.
► Novel methodological work will be completed to compute family co-participation in PA using objective measures.
► The use of mixed methods will provide unique insight and context for our quantitative findings.
► A long-term follow-up (52 weeks postbaseline) will enable assessment of the potential for long-term effects and an accurate assessment of long-term participant retention, which is notoriously challenging in child-based research.
► However, this pilot trial will not allow us to draw conclusions of effectiveness.

## INTRODUCTION

Although the benefits of regular physical activity (PA) in children are well-established,[1 2] roughly only half of UK children meet the recommended 60 min of daily moderate-to-vigorous intensity PA (MVPA).[3] Moreover, observational data reveal that children are less active after school and at weekends compared with during school time, that children in rural areas are less active than their urban counterparts, and that MVPA declines steeply as children enter adolescence, particularly at weekends.[4–7] Intervening prior to adolescence may be important to help young people to reach or maintain adequate PA levels.

Targeting families may, therefore, be one avenue to increase PA among children.[8 9] In particular, parents can influence their children's health behaviours through a variety of mechanisms, including their general parenting style, parenting practices (eg, rule setting, behavioural consequences and

establishing behavioural expectations) and their control over the home environment.[10 11] Parents can also act as gatekeepers to activity[12] and can play an important role in increasing their child's PA through role modelling or parental support.[13] However, little PA promotion research has been conducted in the family compared with the school setting.[14 15] Previous research suggests that: (1) involving family members is critical for sustained behaviour change[13 16 17] and (2) home-based PA interventions that include the family are potentially more effective than those requiring the family to travel to community or other intervention locations.[18 19] Many studies, however, only focus on promoting child PA instead of considering the family as a unit that may work together to change behaviour.[20]

Our recent feasibility study[21] was among the first PA interventions to specifically target whole family engagement. The findings showed that it was feasible and acceptable to deliver and evaluate a family-targeted PA promotion intervention with high acceptability from participating families. Building on this work, this paper describes the protocol for the Families Reporting Every Step to Health (FRESH) pilot trial. The aims of this pilot study are to: assess the feasibility and acceptability of the revised recruitment strategy, intervention and outcome evaluation (after feasibility testing[21]); study long-term retention; and explore further intervention optimisation and preliminary effectiveness.

## METHODS AND ANALYSIS

The Standard Protocol Items: Recommendations for Interventional Trials[22] is used to guide the reporting of this study. We also use the Template for Intervention Description and Replication[23] to guide our description of the intervention.

### Trial design

FRESH is a three-armed, parallel-group, randomised controlled pilot trial using a 1:1:1 allocation ratio with follow-up assessments at 8 and 52 weeks postbaseline (see figure 1). Following baseline assessment, families will be randomly allocated to one of three arms: (1) FRESH arm; (2) pedometer-only arm or (3) no-intervention control arm. All family members in the pedometer-only and FRESH arms receive pedometers and generic family PA promotion information. FRESH families additionally receive access to the intervention website. Control families will be asked to carry on as normal.

### Participants

Families are deemed eligible to participate in this study if there is consent from at least one child in school Years 3–6 (aged 7–11 years, hereafter referred to as the index child) and at least one adult responsible for the index child. The adult must live with the index child in the main household (ie, the index child's primary residence as indicated by the parent). There are no restrictions placed on family type (eg, single parent, shared parenting and inclusion of extended family living in the main household) and no maximum number of participants per family. All participants need to be able to perform light-intensity PA (eg, walking), have access to the Internet and understand the English language sufficiently well to provide informed consent. Family members are able to take part in the intervention irrespective of their participation in

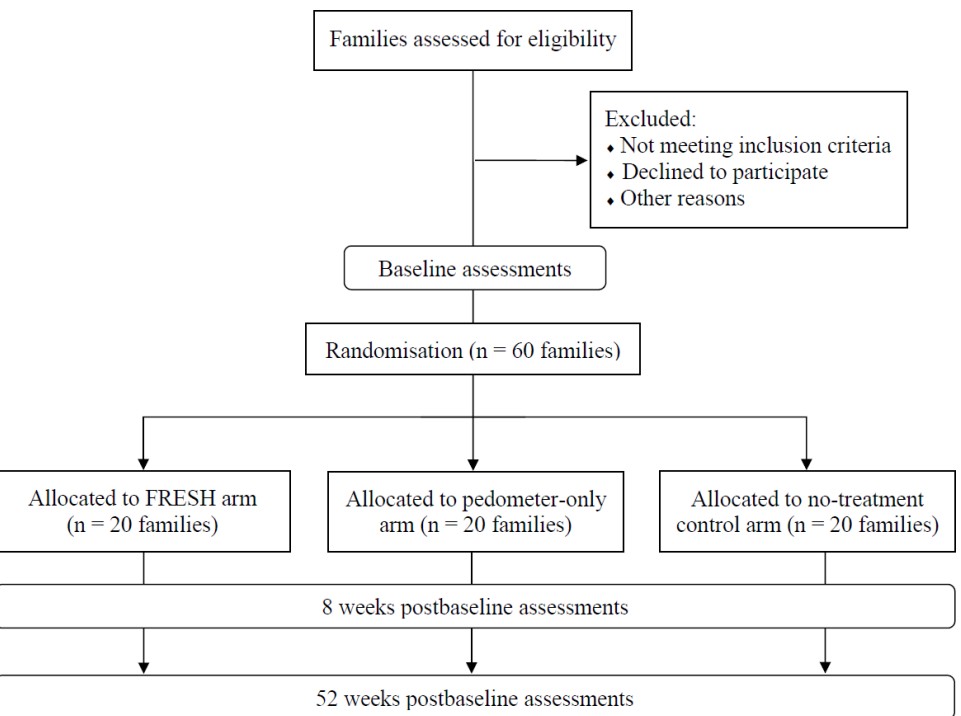

**Figure 1** Participant flow diagram. FRESH, Families Reporting Every Step to Health.

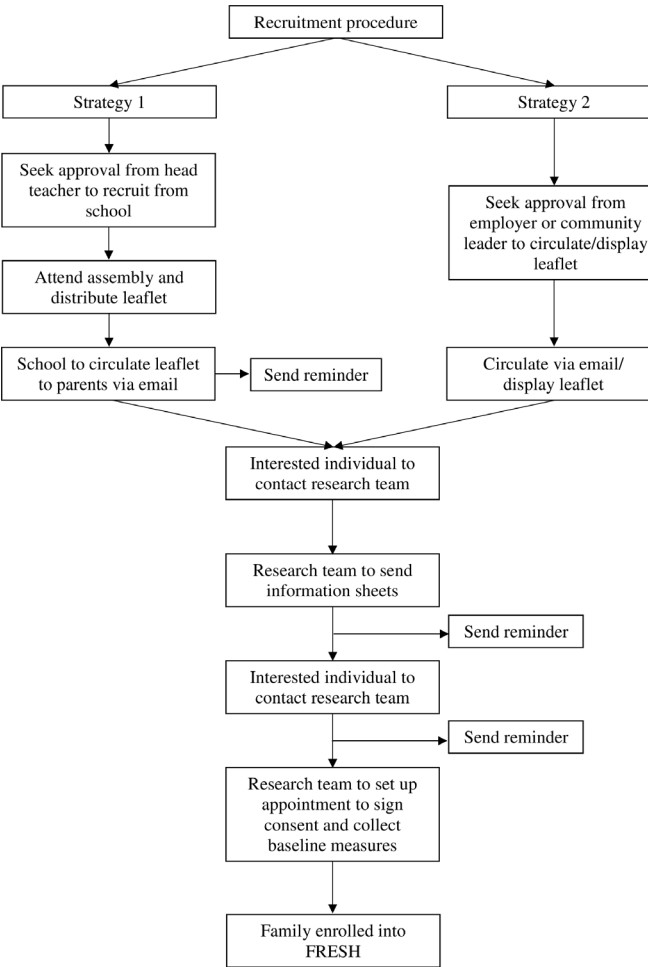

**Figure 2** Recruitment strategy. FRESH, Families Reporting Every Step to Health.

the accompanying evaluation and vice versa. Exclusion criteria related to participation in specific assessments are outlined below with the outcome measures.

### Setting

Families will be recruited from rural Norfolk and Suffolk, counties in East Anglia, UK. Norfolk is 2074 square miles in size and had a total estimated population of 898 400 in 2017 (most recent estimate)[24] and Suffolk is 1466 square miles and had a total estimated population of 756 978 in 2017 (most recent estimate).[25] According to the Norfolk and Suffolk county councils,[26 27] approximately 53% and 42% of the Norfolk and Suffolk populations, respectively, are classified as living in a rural area. Based on the Office for National Statistics[28] classification, 'rural' will be defined as having a postcode falling in a small town, village, hamlet or dispersed settlement. In both counties, existing inequalities have been identified, including PA, obesity and other indicators of child ill health, school readiness and attainment.[6 29 30]

### Recruitment

The recruitment of families is known to be particularly challenging[11 13] and we have described specific recruitment challenges we have encountered previously.[21] To

overcome these challenges, we will use a multifaceted recruitment strategy, including targeting adults (parents) directly and messaging, which focuses on the wider benefits of research participation (eg, spending more time together as a family) as opposed to solely focusing on increasing PA or obesity prevention.[31] Recruitment will be undertaken over an estimated 3 months (with a recruitment rate of ~20 families/month), using two main strategies that target adults and children, as summarised in figure 2. The first strategy involves recruitment in the school setting and the second in employer-based and community-based settings (eg, Brownies/Cubs, community centres and general practitioner surgeries). Alternative recruitment settings may also be explored (eg, online/traditional media) if needed, and will follow the same procedure as the second strategy. For logistical purposes, we aim to find recruitment settings located roughly within an hour commute of Cambridge, UK.

### Recruitment protocol

To recruit schools, employers and community-based organisations, we will aim to first contact those in lead positions (eg, head teachers, human resources, health and well-being leaders, and heads of community-based organisations). An information pack detailing the purpose of the study and all procedures will be included in our correspondence with lead personnel. Also included in the information pack and recruitment material will be a link to a recruitment video, which was developed with families following the suggestion of participants in the FRESH feasibility study.[21] For school-based recruitment, verbal or written approval will be sought to send home study leaflets with children, circulate our leaflet to parents online and send an online reminder to parents approximately 2 weeks later. We will also seek permission to present to Key Stage 2 students (Years 3–6; aged 7–11 years) at a scheduled school assembly. Similarly, for employer-based and community-based recruitment, we will seek approval to circulate our study information to employees or members either online or hardcopy. In all cases, interested parents are asked to express interest by contacting the study team via email or a free-to-call telephone number, after which eligibility will be assessed and study information emailed. A baseline assessment appointment will be made with families still interested in participating. Written informed consent will be obtained for all participating adults and written parental consent and child assent for each participating child during this appointment, prior to baseline assessments.

### Retention

To encourage retention, we will remain in regular contact with all participating families (through intervention website and newsletter/holiday cards), and offer measurement incentives and study feedback. Each individual participant will receive a £5 voucher on return of the accelerometer and global positioning system (GPS) monitors at each measurement time point. Retention will

be monitored by study group and demographic characteristics to observe whether differences in retention occur.

## Randomisation

Randomisation will occur after baseline assessments and the unit of randomisation will be at the family level (ie, the index child and all participating family members). Families will be randomised in blocks of six by an independent statistician using a computer-generated algorithm produced with Stata V.14 and stratified by county (ie, Norfolk or Suffolk). The random allocation sequence will be implemented via a database created in-house on Microsoft Access by independent data management staff. A study coordinator will use the database after baseline to determine which study arm a family is allocated to. No one directly involved in the evaluation will have access to the allocation code or complete sequence.

## Intervention protocol for FRESH arm families

The development, feasibility, acceptability and refinements made to the intervention prior to the current pilot trial have been previously described.[21] In brief, the intervention the FRESH arm will receive is primarily a goal-setting and self-monitoring intervention, delivered online, aimed at increasing PA in whole families. The intervention uses concepts from the socio-ecological model[32] and family systems theory[33] and operationalises constructs from self-determination theory[34] to inform behaviour change strategies. A detailed description of the FRESH intervention components, behaviour change techniques, targeted theoretical constructs and hypothesised mediators is in table 1. Additionally, the FRESH logic model can be found in figure 3.

A week after baseline measures, each family allocated to the FRESH arm will have an hour-long kick-off meeting, scheduled in the family home (or an alternative location can be arranged) with a facilitator. The facilitator will introduce families to the intervention components, accompanying materials (eg, family action planner) and distribute pedometers. Families will also receive their first of four pieces of generic walking information (described in protocol for pedometer-only families section below). However, the main goals of this meeting are to familiarise families with the intervention website and their pedometers and to prompt weekly 'family time' meetings (described in table 1) in which they complete their action planners and select a new challenge city to 'walk to' on the FRESH intervention website. All families will receive a follow-up phone call a week after their kick-off meeting to discuss any issues or ask any clarifying questions. Participant-initiated distant support will continue to be available, where participants can contact the research team with questions or to report issues (eg, website bugs and pedometer issues).

Table 1 provides a detailed description of the FRESH intervention components. In each family, the index child (or children, if multiple) will be designated the role of 'team captain(s)', which involves taking the lead in selecting challenges and uploading steps online. Evidence suggests that children may act as change agents to elicit changes to the psychosocial environment[14]; therefore, promoting the index children to the role of family 'team captain(s)' may strengthen child buy-in, perceived autonomy and improve intervention fidelity. All family members will be given pedometers, wearing them for as long as possible daily, to capture their steps during challenges. Pedometers are simple to use, convenient, and are associated with effective interventions for increasing parent-child PA[35]. After the challenge week is over, whether a family completes their challenge or not, they will receive personalised competence reinforcement messages praising their effort online and hardcopy letters (addressed and mailed to all participating children in the family). In addition, online and tangible rewards will be given to participating children after a challenge week. If a family does not complete a challenge, they will progress to a hidden city along their challenge route, as opposed to the city they chose, and will still receive a reinforcement message and reward, as described. Families with an ongoing challenge will receive email reminders to log steps 3 days and 1 day before an impending challenge ends. After every challenge week, the described cycle will repeat starting with the next 'family time' meeting. Following assessments at 8 weeks postbaseline, families retain access to the website and their pedometers, and can continue using it for as long as they like. There will also be continued support in terms of website updates (eg, leaderboard and parental resource updates) and participants will continue to receive competence reinforcement letters and rewards.

## Protocol for pedometer-only and control families

Following the baseline assessment, families allocated to the pedometer-only arm will be mailed pedometers and generic family PA promotion information produced by Walk4Life, a sub-brand of Change4Life (www.nhs.uk/change4life). Information will continue to be emailed to families (pedometer-only and FRESH arm families) fortnightly on four occasions. The information will provide families with tips to get walking daily and games that can be played while walking. Control families will be asked to carry on as normal and will not receive access to the intervention website, pedometers or any generic information.

## Outcome evaluation measures

Table 2 outlines the measures taken, including assessment order and estimated duration. Data collection will be carried out by two trained research staff and will occur in participating families' homes (or an alternative location by arrangement). Outcomes will be assessed at baseline (prior to randomisation; spring/summer 2018), 8 weeks postbaseline (summer/autumn 2018) and follow-up (52 weeks postbaseline; spring/summer 2019) on all consenting family members (excluding children≤2 years).

**Table 1** Summary of FRESH pilot trial intervention components*

| Intervention components | Dose | Description | Behaviour change techniques | Targeted SDT constructs | Hypothesised mediators |
|---|---|---|---|---|---|
| 1. 'Family time' | Minimum 1×/week, 10–20 min | 'Family time' is expected to provide a weekly (at minimum) opportunity for index children† and family members to review, revise and update their family action planners. Family action planners prompt families to plan PA, monitor weekly steps, and discuss any potential upcoming PA barriers and strategies to overcome them. Index children will be allocated as their family's 'team captain(s)' leading in challenge selection and uploading steps on the FRESH website. | Goal-setting Self-monitoring Positive feedback on progress Social support Praise Positive reinforcement | Perceived competence Perceived relatedness Perceived autonomy | Social support Family social norms for PA PA awareness Basic needs satisfaction PA motivation |
| 2. FRESH website | Minimum 1×/week, 5–20 min | The FRESH website will provide a place for families to self-monitor their step counts and set goals by selecting challenges of varying difficulty. Specifically, the website allows families to 'walk' around the world by choosing one of three target cities to 'walk to' weekly. The challenges are framed as an easy, moderate or hard challenge, which represents a 0%, 5% or 10% increase, respectively, relative to the average steps they take in preceding weeks. Once adults and children accumulate an average of 10 000 and 12 000 steps/day, the step challenge increases will be reduced to 0%, 2.5% and 5%, respectively. Families will also have access to: ▲ A general resources area with suggestions for activities that families could do together. ▲ A map for a visual representation showing the locations families have travelled to. ▲ A step calculator that will convert activities not captured by pedometers to steps (eg, swimming). ▲ Their families' **step history** by challenge. ▲ A **leaderboard**. Families will opt in to being included in a leaderboard that is updated weekly, where families get points for: selecting a challenge; uploading steps; completing a challenge; accumulating a family average of 10 000 steps/ day over the week; increasing steps from the previous week and going on streaks (eg, five completed challenges in a row). Families with an ongoing challenge will receive **email reminders** to log steps 3 days and 1 day before an impending challenge ends. | Goal-setting Self-monitoring Positive feedback on progress Rewards | Perceived competence Perceived relatedness Perceived autonomy | Social support Family social norms for PA PA awareness Basic needs satisfaction PA motivation |
| 3. Pedometry | Throughout intervention (8 weeks) | All family members will receive pedometers to enable self-monitoring and provide immediate feedback. To allow families to view their progress towards their proximal and distal step goals, they will be encouraged to log their steps onto the FRESH website and/or onto the family action planners. | Self-monitoring Immediate feedback | Perceived competence Perceived autonomy | Social support Family social norms for PA PA awareness Basic needs satisfaction PA motivation |

Continued

**Table 1** Continued

| Intervention components | Dose | Description | Behaviour change techniques | Targeted SDT constructs | Hypothesised mediators |
|---|---|---|---|---|---|
| 4. Competence reinforcement/ rewards | ~1×/week (8 weeks) | After completing a challenge or if the challenge week ended, to praise effort (ie, competence reinforcement), children received **personalised supportive letters in the mail** and messages on the FRESH website. They also received small online and **tangible rewards.** ▲ Online rewards will include virtual passport stamps (ie, virtual rewards) and access to reinforcement materials (ie, interactive information about the cities they have walked past during their challenge). ▲ **Tangible rewards** will be in the form of collectable FRESH cards, which display the city names corresponding with the families' challenge. The cards will also enable family to play a card game version of 'rock, paper, scissors'. Children will receive two to four passport stamps/cards for completed challenges (ie, as difficulty increased, more stamps were awarded) and one passport stamp/card for an incomplete challenge. | Feedback on progress Rewards | Perceived competence | Basic needs satisfaction PA awareness |

*This table has been adapted from our feasibility study[21]; bolded text indicates new elements added for the FRESH pilot trial (ie, postfeasibility study).
†The index child refers to the child aged 7–11 years in the family.
FRESH, Families Reporting Every Step to Health; PA, physical activity; SDT, self-determination theory.

## Primary outcome

### PA outcomes

To assess PA, participants will be asked to simultaneously wear an ActiGraph GT3X+ triaxial accelerometer (ActiGraph LLC; Pensacola, Florida, USA) and QStarz Travel Recorder BT1000X GPS monitor (QStarz; Taipei, Taiwan). The accelerometer will be initialised to record data at a sampling rate of 50 Hz, while the GPS will be set to record a location every 10 s. The devices will be worn on each hip during waking hours for 7 consecutive days. A valid week will be defined as a minimum of 420 min/day from 3 days over the 7-day measurement period; however, this definition may be altered depending on participant compliance. Non-wear will be defined as ≥90 min consecutive zeros using the vector magnitude. Raw accelerometer counts will be downloaded and integrated into 5 s epochs. Evenson *et al* cut-points[36] have been recommended to estimate PA intensity in youth[37 38] and Troiano *et al*[39] cut-points for adults; these cut-points will be used in this study. In addition, family co-participation in PA will be measured by matching accelerometer and GPS data using Java. Novel methodological work will be completed to compute minutes of PA per day when family members are together at a given location from the matched data.

## Secondary outcomes

### Health outcomes

Aerobic fitness will be measured using an 8 min submaximal step test (with 2 min rest), which provides an individual calibration of heart rate to work rate (energy expenditure per unit time) to predict a fitness estimate of a participants' heart rate recovery index.[40] Children<7 years will be excluded from the aerobic fitness test.

Height and weight will be measured once with a Leicester portable stadiometer and Seca 877 digital scale, respectively. Waist circumference will be measured twice, using a non-elastic tape measure. A third measure is taken if the first two measures differed by ≥3 cm. Body mass index will be calculated, and converted into age-specific and sex-specific percentiles using standard growth charts for children.[41]

### Behavioural and psychosocial measures

Behavioural and psychosocial measures will be measured via questionnaires distributed to adults and children (those ≤4 years will not complete questionnaires). The questionnaires can be viewed in online supplementary file. These will include: adult and child screen-time use[42–45]; quality of life[46–49]; family co-participation in PA[45]; PA awareness[50 51]; family social norms for PA[52 53]; family support[52]; children's and adult's motivation for PA[54 55]; and children's perceived autonomy, competence and relatedness.[55]

### Family functioning

The Fictional Family Holiday paradigm will be used to assess family functioning via family relationships[56] and connectedness.[57] In this observational paradigm, each

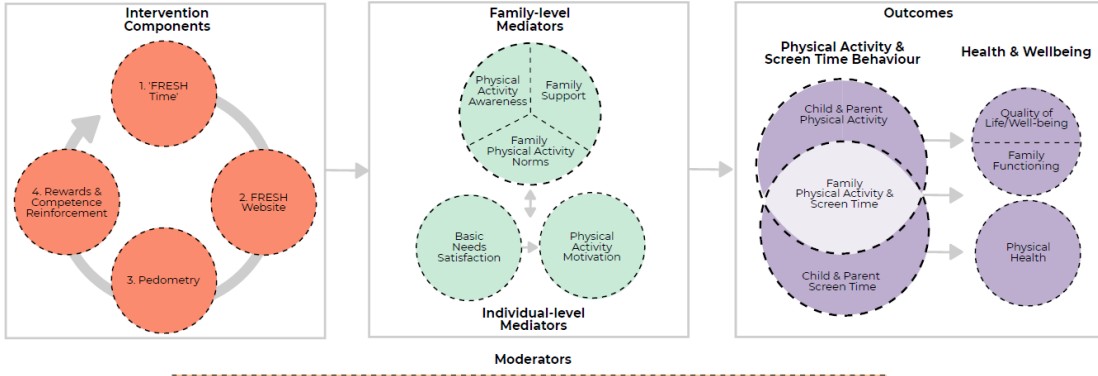

**Figure 3** FRESH theoretical model. FRESH, Families Reporting Every Step to Health.

family is asked to spend 10 min planning and discussing a week-long holiday itinerary with an unlimited budget. The video-recorded activity is then transcribed prior to coding family functioning, which is indexed through multiple conversational markers, including 'power-sharing' (ie, taking turns); parental elicitation of all family members' viewpoints; expressions of individuality (eg, suggestions for destinations/activities or disagreements) and connectedness (eg, agreements, questions or initiating compromise)[57]; as well as non-verbal markers, including expressions of amity (eg, laughter and displays of affection) or hostility (eg, sarcasm and anger).

**Table 2** Order of measures and estimated duration of data collection*†

| Measure | Duration |
| --- | --- |
| 1. Anthropometric measures (height, weight and waist circumference) | 5 min/person |
| 2. Questionnaire | 20 min/family |
| 3. Blood pressure‡ | 10 min/person |
| 4. Step test (aerobic fitness) | Prep: 5 min/person Test: 10 min/family |
| 5. Accelerometer and GPS explanation | 5 min/family |
| 6. Fictional Family Holiday (family functioning) | 10 min/family |
| **Total duration of baseline measurement session (includes consent/assent process)** | **120 min** |
| **Total duration of subsequent measurement sessions** | **105 min** |

*This table has been adapted from our feasibility study.
†Estimate based on a four-person household, total time increases by ~30 min per additional family member.
‡Duration is halved when calculating total duration because multiple monitors will be used to enable two family members to be measured simultaneously.
GPS, global positioning system.

### Family out-of-pocket expenditure for PA
PA-related expenditure for each family member will be collected via a questionnaire that was developed and refined following our feasibility study.[21] The questionnaire comprises two questions about expenditure related to membership fees and subscriptions (eg, for sports clubs and fitness centres) and sports equipment (eg, sportswear and gadgets) and is completed by the same adult at each time point for their whole family.

### Process evaluation
A mixed methods process evaluation will be conducted after assessments 8 weeks postbaseline. Using opened-ended and 4-point Likert-scale questions (1=strongly disagree and 4=strongly agree), adults will self-report their overall opinion of FRESH, the intervention components, measurements and suggestions for improvement. Children will also self-report on the described topics, responding to dichotomous 'yes/no' questions. In addition, semistructured focus groups will be conducted online on 10/20 FRESH families and 5/20 families from each of the other two arms (ie, pedometer-only and control). Focus groups will focus on families' experience taking part in the trial, perceived acceptability of individual intervention components, intervention fidelity, challenges/barriers encountered and suggested improvements, as appropriate based on study arm allocation. All focus groups will be transcribed verbatim. We will also explore FRESH families' engagement with the intervention website (eg, page views and challenges accepted/completed) and aspects of the recruitment process (eg, recruitment duration, resources used and comparisons of recruitment strategies).

### Patient and public involvement
FRESH was developed with substantial input from children and families from the public, which has been described elsewhere.[21] Since the completion of the FRESH feasibility study, families from the public have been further involved with the optimisation of FRESH in a number of ways. As mentioned in section Recruitment protocol, we sought the involvement of families

from the public to develop a recruitment video; these families helped to develop the script and acted in the video, which can be viewed online (www.youtube.com/watch?v=UxUHN1JsjUM). We also asked families to engage with the FRESH website and provide feedback to inform modifications that could be made. Lastly, we have included parents as members on our Study Steering Committee.

### Blinding

Researchers conducting analyses will be blinded to family treatment allocation. Research assistants will not be explicitly told family treatment allocation. However, it will not be possible to conceal treatment allocation from participating families.

### Sample size considerations

Since this is a pilot study, a sample size calculation will not be performed. We plan to recruit a sample of 60 families, with a sample size of ~180–240 participants based on ~3–4 members per family. Our estimated sample size is based on prior study experience[31] and sample sizes of previous pilot studies.[58 59]

### Data analysis
#### Quantitative data

Statistical analyses of the primary and applicable secondary outcomes will be conducted using linear mixed models in Stata V.14. The models will be used to assess the impact of treatment (FRESH, pedometer-only or control), time, (baseline, 8 weeks and 52 weeks) and the group-by-time interaction. An estimate of effect and 95% CI will be calculated. Descriptive statistics will also be calculated to describe data related to recruitment, retention, acceptability, family functioning and website engagement.

#### Economic analyses

Resource use counts (eg, time spent training families) will be converted to cost using unit costs from a common price year, and adjusted to the common price year using the consumer price index. Total cost per family will be the sum of intervention delivery and PA expenses in each arm. Incremental cost per family at each time point will be combined with a change in MVPA to calculate a measure of cost-effectiveness. Analysis of uncertainty will include reporting 95% CIs around increments and the cost-effectiveness acceptability curve, showing the probability of cost-effectiveness as a function of willingness to pay for an hour of MVPA (taking account of dominance and extended dominance as appropriate). The incremental cost per quality-adjusted life year (QALY) of the FRESH interventions over a 10-year horizon will also be calculated. The emphasis of these analyses will not be on the point estimate means, but on identifying the uncertainty in cost-effectiveness, informing a value of information analysis.[60] The value of information analysis will be conducted to predict the efficient sample size of the definitive FRESH trial as a function of *willingness to pay*

*for an additional hour of MVPA,* and using the modelled results, predict the efficient sample size as a function of *willingness to pay* for a QALY.

#### Qualitative data

A content analysis will be conducted using existing guidelines[61] to explore the feasibility and acceptability of the revised FRESH intervention, outcome evaluation and suggestions for further intervention optimisation via family focus groups. Specifically, the analysis will be conducted in two separate phases. During the data organisation phase, text from each transcript will be divided into segments (meaning units) to produce a set of concepts that reflected meaningful pieces of information.[61] Tags will then be assigned to each meaning unit. Tagging will be performed by one researcher, with a second double-tagging ~25% of transcripts. For the data interpretation phase, the inventory of tags from all transcripts will be examined by two researchers, which will lead to the emergence of themes and subthemes within each overarching category.

## DISCUSSION

The FRESH study will provide a response to calls for the need for innovative interventions targeting young people and families[62] and builds on our previous work.[21] As far as we are aware, FRESH is among the first PA interventions that specifically targets and evaluates whole family engagement.

The overall aim of a future definitive trial will be to establish the long-term effectiveness and cost-effectiveness of the family-based FRESH intervention to promote MVPA in young people and their families. A decision to progress to a definitive trial will be made with an independent Study Steering Committee, using the following parameters and taking into account qualitative findings on the acceptability of trial procedures:

► Demonstrable feasibility of recruiting 20 families/month (accounting for increased staffing in a future definitive trial) and retaining 75% of index children at follow-up (52 weeks postbaseline).
► Good intervention adherence for families in the FRESH arm, defined as >75% of families uploading steps at least six times in the first 3 months of the study.
► Intervention optimisation is feasible (ie, identified adaptations are practical, affordable and acceptable).
► Evidence to suggest an adequately powered trial would require a feasible number of participants.
► Discontinuation of trial arm based on evidence of harm or limited acceptability/feasibility.
► Positive expected net gain of sampling from a definitive trial.

### Ethics and dissemination

Written informed consent will be obtained for all participating adults and written parental consent and child assent for each participating child prior to collecting baseline assessments.

**Acknowledgements** This work was undertaken by the Centre for Diet and Activity Research (CEDAR), where funding from the Cancer Research UK, the British Heart Foundation, the Economic and Social Research Council, the Medical Research Council, the National Institute for Health Research and the Wellcome Trust, under the auspices of the UK Clinical Research Collaboration, is gratefully acknowledged (087636/Z/08/Z, ES/G007462/1 and MR/K023187/1).

**Contributors** EvS (principal investigator), HEB, CH, APJ and KLM secured funding for the research. All the authors contributed to the study design. JMG drafted the manuscript. HEB, EC, ESH, CH, APJ, KLM and EvS critically reviewed and revised the manuscript. All the authors read and approved the final manuscript.

**Funding** This work was supported by the National Institute for Health Research Public Health Research Programme (project number 15/01/19). Intervention costs for the current study were supported by Active Norfolk and Suffolk County Council. Funding was also received from the Medical Research Council (project number MC_UU_12015/7).

**Competing interests** None declared.

**Patient consent for publication** Not required.

**Ethics approval** Ethical approval has been obtained from the Ethics Committee for the School of the Humanities and Social Sciences at the University of Cambridge (ID number: 17/113).

**Provenance and peer review** Not commissioned; externally peer reviewed.

**ORCID iD**
Justin M Guagliano http://orcid.org/0000-0002-4450-5700

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
