## [Reviewer comments · BMJ Open]

ARTICLE DETAILS

TITLE (PROVISIONAL)	A whole family-based physical activity promotion intervention: The Families Reporting Every Step to Health (FRESH) pilot randomised controlled trial protocol
AUTHORS	Guagliano, Justin M; Brown, Helen Elizabeth; Coombes, Emma; Haines, Elizabeth S; Hughes, Claire; Jones, Andrew P; Morton, Katie L; van Sluijs, Esther

VERSION 1 – REVIEW

REVIEWER	Sally Kerry Queen Mary University of London, UK
REVIEW RETURNED	03-May-2019

GENERAL COMMENTS	This is a well designed pilot study with appropriate methods of analysis. I have 3 suggestions for improvement of greater clarity that do not affect the basis design. The journal editor recommends that dates are given for all protocol and none are given here. Recruitment targets are 20 /month. However the recruitment process is long and so it isn't clear what this means. It would also be useful to measure the resources used in some way and be able to compare the different strategies. It would also be useful to have a timetable for recruitment and to monitor how long each step takes, how much staff time is involved and estimate how it would be scaled up. One of the objectives is 'Discontinuation of trial arm based on evidence of harm or limited acceptability/feasibility;' Should this simply be limited acceptability because it is not clear than harms are actually being measured or what 'harms' are likely to mean in this context.
---

REVIEWER	Dr Gisela Nyberg Karolinska Institutet, Sweden
REVIEW RETURNED	30-May-2019

GENERAL COMMENTS	Manuscript ID bmjopen-2019-030902, entitled "A whole family-based physical activity promotion intervention: The Families Reporting Every Step to Health (FRESH) pilot randomised controlled trial protocol." I believe that the study is very important since family interventions are greatly needed in order to increase physical activity in children and few family intervention studies have been conducted. Please find my comments below.
--

	The manuscript is very well written, perfectly structured and the intervention is very well described guided by the SPIRIT and the Template for Intervention Description and Replication My main concern with the intervention is that the intervention period is very short, only 8 weeks. I do not think that the duration is long enough in order to promote a changed behaviour. Many earlier studies show short lived effects after physical activity interventions and effects are not maintained at a later follow-up. Therefore, many studies use “boosters” after the intervention in order to remind participants about maintaining their changed lifestyle behaviours. What is the rationale for the short intervention period and have they thought about “boosters”? Another concern is how it will be possible to scale up the intervention after the pilot study. In the pilot study the families have individual meetings and data collection in the homes. How will that be possible in a larger study? Why do they plan to have a three-armed intervention? Wouldn't it be better to only have two, the FRESH arm and the control group? Please, provide the rationale for inclusion of participants from rural areas in the background. Can you explain the reason for including families with at least one child that is in the age of 7-11? What is meant by the main household, does it work if the child live part time at both their parents? As mentioned in the manuscript (page 6, line 128), it is challenging to recruit families to interventions. I think that there might be a problem to keep the control families in the study if they are not offered anything in return. Why not offer them the intervention after they have been controls? What will happen if the parents decide not to participate and the child still wants to take part in the intervention? In the FRESH intervention arm: will there be separate meetings for each family in their own homes? How will you precede if families do not want to invite the researcher to their own homes? If there are more than one child in the target age, who will be the "team captain"? Is there any risk that the information about the study, that you intend to give to potential participants during the recruitment will affect their perceptions about the study in a positive direction? Outcome evaluation: On page 10, the behavioural and psychosocial measures are described. References are shown for different outcomes but not the specific questions. Is it possible to specify the questions or put them in the appendices? In Figure 3 the theoretical model and the mediators are shown. Are the authors planning to conduct any mediation analyses? Some more information is needed about how they are planning to conduct the cost-effectiveness analyses.
--	---

VERSION 1 – AUTHOR RESPONSE

Reviewer 1 comments	Authors' responses
1. This is a well-designed pilot study with appropriate methods of analysis. I have 3 suggestions for improvement of greater clarity that do not affect the basis design.	We would like to thank the Reviewer for their positive words and thoughtful review of our manuscript. Below we have addressed the Reviewer's specific comments.
2. The journal editor recommends that dates are given for all protocol and none are given here.	We have now added dates for data collection on page 10, lines 230-233: “Outcomes will be assessed at baseline (prior to randomisation; spring/summer 2018), 8 weeks post-baseline (summer/autumn 2018), and follow up (52 weeks post-baseline; spring/summer 2019) on all consenting family members (excluding children ≤ 2 years).”
3. Recruitment targets are 20 /month. However the recruitment process is long and so it isn't clear what this means. It would also be useful to measure the resources used in some way and be able to compare the different strategies. It would also be useful to have a timetable for recruitment and to monitor how long each step takes, how much staff time is involved and estimate how it would be scaled up.	As stated on pages 14-15, lines 354-369, we have outlined specific criteria that should be achieved in order to progress to a definitive trial. One of the criteria is to demonstrate the feasibility of a recruitment rate of 20 families/month (accounting for increased staffing in future definitive trial). For clarity, however, we have added the recruitment rate and timetable for recruitment under the 'Recruitment' heading: “Recruitment will be undertaken over an estimated 3 months (with a recruitment rate of ~20 families/month) using two main strategies that target adults and children, as summarised in Figure 2.” Page 6, lines 134-136. We appreciate the Reviewer's suggestion regarding exploring aspects of the recruitment process in greater detail and have added this to recommendation to the manuscript: “We will also explore FRESH families' engagement with the intervention website (e.g., page views, challenges accepted/completed) and aspects of the recruitment process (e.g., recruitment duration, resources used, comparisons of recruitment strategies).” Page 12, lines 293-295.
4. One of the objectives is 'Discontinuation of trial arm based on evidence of harm or limited acceptability/feasibility;' Should this simply be limited acceptability because it is not clear that harms are actually being	The discontinuation of a trial arm based on evidence of harm or limited acceptability/feasibility is one of the progression criteria (not an objective) that we have outlined

measured or what 'harms' are likely to mean in this context.	to inform whether progression to a definitive trial is warranted. This criterion was included in our grant funded by the National Institute for Health Research Public Health Research Programme (project number 15/01/19); therefore we cannot change the wording of this criterion. Regarding the Reviewer's comment about harms, we are not measuring harms, but rather monitoring harms (e.g., adverse events, disbenefit) monthly.
Reviewer 2 comments	Authors' responses
1. I believe that the study is very important since family interventions are greatly needed in order to increase physical activity in children and few family intervention studies have been conducted. Please find my comments below.	We appreciate the Reviewer's positive comment and critical review of our manuscript. We believe our manuscript is stronger thanks to both Reviewers' comments. We have addressed the Reviewer's specific comments below, where possible.
2. The manuscript is very well written, perfectly structured and the intervention is very well described guided by the SPIRIT and the Template for Intervention Description and Replication. My main concern with the intervention is that the intervention period is very short, only 8 weeks. I do not think that the duration is long enough in order to promote a changed behaviour. Many earlier studies show short lived effects after physical activity interventions and effects are not maintained at a later follow-up. Therefore, many studies use "boosters" after the intervention in order to remind participants about maintaining their changed lifestyle behaviours. What is the rationale for the short intervention period and have they thought about "boosters"?	Thank you for the compliment about the structure and writing of our manuscript. Regarding the comment about an 8-week intervention duration being too short to promote behaviour change, as we state on page 9 (lines 212-214), families will retain access to the website, their pedometers, and can continue using them for as long as they like after 8-week post-baseline assessments. There will also be continued support in terms of website updates (e.g., leaderboard and parental resource updates) and participants will continue to receive competence reinforcement letters and rewards. The latter detail was added on page 9, lines 215-217). In addition, we would like to reiterate that the purpose of this pilot trial is primarily to assess the feasibility and acceptability of the revised recruitment strategy, long-term retention, FRESH intervention and outcome evaluation – not the effect of the intervention on hypothesised outcomes. Further, none of the predefined progression criteria we have set are dependent upon demonstrating effectiveness, we have mainly included that we would explore preliminary effectiveness to inform sample size calculations should we progress to a definitive trial in future.
3. Another concern is how it will be possible to scale up the intervention after the pilot study. In the pilot study the families have	There appears to be two questions here, one relates to scaling up intervention delivery and the other relates to scaling up data collection.

individual meetings and data collection in the homes. How will that be possible in a larger study?	Intervention delivery can rather easily be scaled up as our intervention is delivered online and the individual 'kick-off' meeting can also be delivered online. In terms of scaling up data collection, this will also be possible by increasing research assistant staffing and possibly a multi-centre trial, which would be accounted for in a grant proposal budget.
4. Why do they plan to have a three-armed intervention? Wouldn't it be better to only have two, the FRESH arm and the control group?	Given that we are still in the pilot phase of this trial, we have the opportunity to take a slightly more exploratory approach than perhaps we might take during a definitive trial. Therefore, we decided to include three arms in this pilot trial to explore if feasibility/acceptability and preliminary effectiveness of the FRESH arm is greater than the other two arms. The inclusion of the pedometer-only arm provides us with additional insights into the value of the FRESH intervention over and above pedometers alone.
5. Please, provide the rationale for inclusion of participants from rural areas in the background.	We have added rationale for the inclusion of participants in rural areas in the background, the sentence now reads: "...observational data reveal that children are less active after school and at weekends compared to during school time, that children in rural areas are less active than their urban counterparts, and that MVPA declines steeply as children enter adolescence, particularly at weekends [4-7]." Page 4, lines 65-68.
6. Can you explain the reason for including families with at least one child that is in the age of 7-11?	As we state in our Introduction, moderate-to-vigorous intensity physical activity (MVPA) declines sharply as children enter adolescence (page 4, lines 63-69). Therefore, intervening prior to the steep decline observed in adolescence may be important to help young people reach or maintain adequate physical activity levels – we have added this sentence to the manuscript on page 4, lines 68-69.
7. What is meant by the main household, does it work if the child lives part time at both their parents?	We used the term 'main household' to mean the index child's primary residence. To be deemed eligible to participate in this study, at least one 7-11 year old index child and one adult living in the main household with the index child were required. However, no other restrictions were placed on family type. For example, shared parenting families were eligible for participation as long as one adult living in the main household was willing to participate. We have

	amended the manuscript for clarity on the two points above, it now reads: “The adult must live with the index child in the main household (i.e., the index child’s primary residence as indicated by the parent). There are no restrictions placed on family type (e.g., single parent, shared parenting, inclusion of extended family living in the main household) and no maximum number of participants per family.” Page 5, lines 108-111.
8. As mentioned in the manuscript (page 6, line 128), it is challenging to recruit families to interventions. I think that there might be a problem to keep the control families in the study if they are not offered anything in return. Why not offer them the intervention after they have been controls?	We acknowledge that it is possible that retention may be an issue; however, one of the reasons for conducting this trial is to assess the long-term retention of families. We chose not to offer control families the intervention because we anticipate further intervention optimisation may be required (e.g., website optimisation).
9. What will happen if the parents decide not to participate and the child still wants to take part in the intervention?	To be deemed eligible to participate in this study, at least one 7-11 year old index child and one adult living in the main household with the index child were required (stated on page 5, lines 106-108.). This is to ensure that we can assess the family-based secondary outcomes (e.g. family functioning and family co-participation in physical activity). However, as we mention on page 5, lines 113-115, additional family members are able to take part in the intervention irrespective of their participation in the accompanying evaluation and vice versa. In the case highlighted by the reviewer, if both parents did not provide consent for the measurements, the family would not take part.
10. In the FRESH intervention arm: will there be separate meetings for each family in their own homes? How will you precede if families do not want to invite the researcher to their own homes? If there are more than one child in the target age, who will be the "team captain"?	The ‘kick-off’ meetings for FRESH arm families are conducted separately by family and alternative arrangements can be made if families are uncomfortable having members of the research team in their home (for the ‘kick-off’ meeting or data collection), we have made these details clearer in the manuscript: “A week after baseline measures, each family allocated to the FRESH arm will have an hour-long kick-off meeting, scheduled in the family home (or an alternative location can be arranged) with a facilitator.” page 8, lines 184-186.

	“Data collection will be carried out by two trained research staff and will occur in participating families’ homes (or an alternative location by arrangement).” pages 9-10, lines 228-230. If multiple children were eligible to be their family’s ‘team captain’, we designated them as co-captains. We have added this detail to page 8, lines 196-201, it now reads: “In each family, the index child (or children, if multiple) will be designated the role of ‘team captain(s)’, which involves taking the lead in selecting challenges and uploading steps online. Evidence suggests that children may act as change agents to elicit changes to the psychosocial environment [14]; therefore, promoting the index children to the role of family ‘team captain(s)’ may strengthen child buy-in, perceived autonomy, and improve intervention fidelity.”
11. Is there any risk that the information about the study, that you intend to give to potential participants during the recruitment will affect their perceptions about the study in a positive direction?	As with all studies, our ethical approval requires that we provide all participants with full information about the study procedures before obtaining consent. The purpose of any recruitment material for any study is to attract participants to participate in the advertised study, so one should hope their recruitment material is perceived positively by those exposed to the material. However, all potential participants exposed to FRESH recruitment material, regardless of recruitment strategy, were exposed to the same recruitment material (i.e., recruitment video, leaflets). The material explained what participation in FRESH would involve in broad terms, but no specific information about the intervention was divulged.
12. Outcome evaluation: on page 10, the behavioural and psychosocial measures are described. References are shown for different outcomes but not the specific questions. Is it possible to specify the questions or put them in the appendices?	We have added the parent and child questionnaires used to assess behavioural and psychosocial outcomes in a Supplementary File.
13. In Figure 3 the theoretical model and the mediators are shown. Are the authors planning to conduct any mediation analyses?	We have hypothesised mediators in our theoretical model; however, we do not have plans to conduct mediation analyses for this phase of testing as our sample size for this pilot

	study will not be sufficient to assess effectiveness or study mediation.
14. Some more information is needed about how they are planning to conduct the cost-effectiveness analyses.	We have added more detail regarding our planned economic analyses on page 13, lines 325-338, which reads: “Economic analyses. Resource use counts (e.g. time spent training families) will be converted to cost using unit costs from a common price year, and adjusted to the common price year using the consumer price index. Total cost per family will be the sum of intervention delivery and physical activity expenses in each arm. Incremental cost per family at each time point will be combined with change in MVPA to calculate a measure of cost-effectiveness. Analysis of uncertainty will include reporting 95% confidence intervals around increments and the cost-effectiveness acceptability curve, showing the probability of cost-effectiveness as a function of willingness to pay for an hour of MVPA (taking account of dominance and extended dominance as appropriate). The incremental cost per quality-adjusted life year (QALY) of the FRESH interventions over a 10 year horizon will also be calculated. The emphasis of these analyses will not be on the point estimate means, but on identifying the uncertainty in cost-effectiveness, informing a value of information analysis [60]. The value of information analysis will be conducted to predict the efficient sample size of the definitive FRESH trial as a function of willingness to pay for an additional hour of MVPA, and, using the modelled results, predict the efficient sample size as a function of willingness to pay for a QALY.”

VERSION 2 – REVIEW

REVIEWER	Dr Gisela Nyberg Karolinska Institutet Department of Public Health Sciences 171 77 Stockholm Sweden
REVIEW RETURNED	14-Sep-2019
GENERAL COMMENTS	Thank you for addressing all the comments. I believe that the manuscript is much clearer now and I am happy with the response.